# *Can you Summarize my learnings?* Towards Multi-modal Perspective-based Educational Dialogue Summarization

**Raghav Jain**[1], **Tulika Saha**[2], **Jhagrut Lalwani**[3], and **Sriparna Saha**[1]

[1]Dept. of CSE, Indian Institute of Technology Patna, India
[2] University of Liverpool, United Kingdom
[3] Veermata Jijabai Technological Institute, India
raghavjain106@gmail.com, sahatulika15@gmail.com

## Abstract

The steady increase in the utilization of Virtual Tutors (VT) over recent years has allowed for a more efficient, personalized, and interactive AI-based learning experiences. A vital aspect in these educational chatbots is summarizing the conversations between the VT and the students, as it is critical in consolidating learning points and monitoring progress. However, the approach to summarization should be tailored according to the *perspective*. Summarization from the VTs perspective should emphasize on its teaching efficiency and potential improvements. Conversely, student-oriented summaries should distill learning points, track progress, and suggest scope for improvements. Based on this hypothesis, in this work, we propose a new task of *Multi-modal Perspective based Dialogue Summarization (MM-PerSumm)*, demonstrated in an educational setting. Towards this aim, we introduce a novel dataset, *CIMA-Summ* that summarizes educational dialogues from three unique perspectives: the Student, the Tutor, and a Generic viewpoint. In addition, we propose an *Image and Perspective-guided Dialogue Summarization (IP-Summ)* model which is a Seq2Seq language model incorporating **(i)** multi-modal learning from images and **(ii)** a perspective-based encoder that constructs a dialogue graph capturing the intentions and actions of both the VT and the student, enabling the summarization of a dialogue from diverse perspectives. Lastly, we conduct detailed analyses of our model's performance, highlighting the aspects that could lead to optimal modeling of *IP-Summ*.

## 1 Introduction

Artificial Intelligence (AI) is making significant strides in the field of education, paving the way for numerous innovative applications. The power of Natural Language Processing (NLP) and advanced machine learning techniques have given rise to Virtual Tutors (VTs) (Stasaski et al., 2020; Jain et al.,

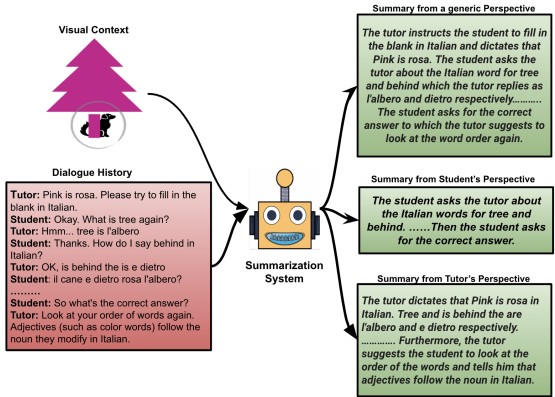

Figure 1: Sample instance of *MM-PerSumm* task

2022), which, unlike traditional methods, are accessible to anyone, anywhere, anytime[1]. VTs are revolutionizing the education sector, breaching the barriers of geographical boundaries, and democratizing learning on a global scale. They offer a tailored learning approach, adapting to each learner's pace, aptitude, and preferences, thereby catering to a diverse array of learning styles. While the implementation of AI in education are evident, extracting valuable insights from these virtual educational interactions remains a challenging task.

Summarizing insights from VT carries significant implications for both the improvement of AI-based educational systems and the learning outcome of students. With a growing reliance on these AI-powered tools for teaching and learning, it becomes crucial to gauge their teaching efficiency, highlight potential areas of improvement, and assess their overall impact on learners' progress. For learners, summaries provide a distillation of key learning points, tracking their learning journey and suggesting areas for future study, which could be pivotal in informing their learning path. While dialogue summarization techniques (Feng et al., 2022) have seen extensive development in several domains, their application to educational interactions, particularly with VTs, has not been thor-

---

[1]https://tech.ed.gov/files/2017/01/NETP17.pdf

oughly explored. Furthermore, the importance of summarizing these dialogues from multiple *perspectives* - those of the tutor, the student, and a neutral observer - is a distinct requirement in this domain (Jain et al., 2022). These perspectives can reveal diverse, valuable insights that can contribute to the evolution of virtual tutoring systems and the individual learning journey of students. Thus, this area holds vast potential for exploration and advancement in the field of AI-driven education.

To address this significant gap in the field, the **contributions** of this paper are : **(i)** Propose a new task of *Multi-modal Perspective based Dialogue Summarization (**MM-PerSumm**)* which is demonstrated in an educational setting; **(ii)** As a step towards this goal, we extend an existing education based conversational dataset, *CIMA* (Stasaski et al., 2020) by summarizing the dialogues from three different perspectives: that of the student, the tutor, and a generic viewpoint to create **CIMA-Summ** which serves as the foundation for our research; **(iii)** To address the task of *MM-PerSumm*, we propose *Image and Perspective-guided Dialogue Summarization* (**IP-Summ**) model which leverages a Seq2Seq language model with a perspective-based encoder. It is designed to construct a dialogue graph, capturing the intentions and actions of both tutors and students, thereby enabling the summarization of educational dialogues from diverse perspectives. By incorporating multi-modal learning in the form of image-based instruction as a contextual cue, our approach allows the *IP-Summ* to leverage both textual and visual information for more comprehensive and accurate summarization.

## 2 Related Works

In this section, we detail some relevant works in the context of Educational-NLP and Dialogue Summarization in general.

**Education and AI.** With the advent of AI, tailored educational materials can be generated based on individual learning styles and abilities (Bhutoria, 2022). Automated feedback systems offer timely evaluations of assignments and homework (Kumar et al., 2022; Filighera et al., 2022). Educational question generation is another area that has witnessed significant advancements (Elkins et al., 2023). In the context of virtual tutoring, advancements in NLG have paved the way for interactive and immersive learning experiences. This led to the creation of AI-related datasets, such as CIMA (Stasaski et al., 2020) and the Teacher-Student Cha-

troom Corpus (Caines et al., 2022). Recently, a lot of attempts have been made to create a VT system utilizing these datasets (Jain et al., 2022; Macina et al., 2023).

**Dialogue Understanding.** In the context of the Social Good theme, there have been an upsurge of research in the latest time focused on understanding dialogues. For example works in mental health conversations (Saha et al., 2022a,b,c, 2021), disease diagnosis assistant (Tiwari et al., 2022; Saha et al., 2023), education (Jain et al., 2022) etc. A broad spectrum of research has been done on dialogue summarization spanning different domains. One of the notable sectors includes meeting summarization, where advancements have been observed in both extractive and abstractive techniques (Wang, 2022; Rennard et al., 2023). Similarly, chat summarization has been a focus due to the explosion of messaging apps and business communication platforms (Gliwa et al., 2019; Feng et al., 2021). Email thread summarization (Carenini et al., 2007) is another domain that has seen significant research interest. The complexity of the task lies in identifying the salient information buried in long and often nested conversations (Mukherjee et al., 2020). The domain of customer service summarization (Liu et al., 2019) has a more specific goal, which is to extract customer issues. Finally, medical dialogue summarization (Joshi et al., 2020; Song et al., 2020; Enarvi et al., 2020; Wei et al., 2023) has seen substantial development, driven by the need for concise patient-doctor conversation records.

## 3 Dataset

To facilitate research in educational dialogue summarization, we create the *CIMA-Summ* dataset, which is based on an existing conversational dataset for educational tutoring.

### 3.1 Data Collection

This study builds upon the existing CIMA dataset (Stasaski et al., 2020), a valuable resource in the field of dialogue systems, particularly in the context of tutoring dialogues. This dataset was chosen because it is uniquely positioned in the intersection of dialogue systems and education while also being supplemented with a rich set of features like object images and intent-action labels. CIMA dataset contains one-to-one student-tutor conversations where the aim is to assist students in learning the Italian translation of an object and its characteristics. Each object discussed within the dialogue

is also supplemented with a corresponding image. Each student utterance contains intent tags, namely, *Guess*, *Question*, *Affirmation*, or *Other*, while the Tutor's action are categorized as *Question*, *Hint*, *Correction*, *Confirmation*, and *Other*. However, in its original form, the CIMA dataset provides the intent-action labels only for the concluding student utterance and the corresponding gold tutor response for each dialogue. To overcome this limitation, we draw upon the silver action-intent labels from the *extended-CIMA* dataset (Jain et al., 2022).

## 3.2 Data Annotation

A detailed annotation approach was followed which is discussed below.

**Annotation Guidelines.** A clear set of annotation guidelines was established which were aimed at providing a structured way for annotators to distill key points from the dialogues and summarize them from three distinct perspectives: the student, the tutor, and an overall dialogue summary.

**Student Perspective :** Annotators were tasked with crafting a summary that encapsulates the key takeaways for the student from the dialogue. This includes the new knowledge gained, any misconceptions corrected, and the overall progress in understanding. This summary should reflect what the student has learned during the interaction.

**Tutor Perspective :** For the tutor perspective, annotators were instructed to focus on the teaching strategies employed by the tutor, the clarification of student doubts, and the overall guidance provided. The summary should highlight the tutor's effort in facilitating the student's learning.

**Overall Perspective :** This is aimed at presenting a balanced view of the conversation both from the student's and tutor's perspective, highlighting key dialogic exchanges, instructional elements, and learning outcomes. *More details on the annotation guidelines is presented in the Appendix section.*

**Annotation Process.** We hired three annotators from the author's affiliation who were graduates in English Linguistics with substantial familiarity in the educational/acaedemic background for our annotation task. The annotators were first trained to summarise dialogues using a set of pre-annotated gold-standard samples from the *CIMA-Summ* dataset. These examples were annotated with three types of summaries (student perspective, tutor perspective, and overall dialogue summary) by two experienced researchers (from author's collaboration) specializing in educational dialogue systems. The intention was to provide a diverse set of examples that demonstrated a range of topics and dialogue scenarios, giving the annotators an understanding of the depth and breadth of the task. Feedback sessions were arranged where the annotators interacted with the experienced researchers to discuss the evaluations and ways to enhance the quality of the summaries. *More details on how the annotators were trained is detailed in the Appendix section.* Finally, the main annotation task was conducted using the open-source platform Doccano[2], deployed on a Heroku instance. Given the intricacy of the task and the need to ensure a high-quality annotation, we followed a structured schedule over a span of five days in a week (*the details of which are mentioned in the Appendix section*). This annotation process took approximately five weeks to complete. The final annotations demonstrated a high level of quality, with average fluency and adequacy scores of 4.98 and 4.87, respectively. Furthermore, the evaluators showed high inter-agreement levels, with unanimous scores of 94.7% for fluency and 89.1% for adequacy ratings. *More details on the annotation quality assessment is detailed in the Appendix section.*

## 3.3 CIMA-Summarization : *CIMA-Summ Dataset*

The *CIMA-Summ* dataset comprises of 1134 dialogues with each dialogue annotated with three different perspective amounting to a total of 3402 parallel summaries accompanied with visual cues. A sample instance from the *CIMA-Summ* dataset is shown in Figure 2.

**Qualitative Analysis.** Figure 3(a) shows the distribution of word count for three type of summaries. The median length of the student's summary is less than the tutor's summary, which in turn is less than the overall summary. Tutor might tend to use more technical language in the interaction to accurately convey the information, which can contribute to longer summaries compared to students, who may only focus on key points. Overall summary, encompassing both tutor's and student's perspectives, is bound to have the highest median length amongst the three. Figure 3(c) shows the distribution of similarity for pairs of summaries. It is evident that the similarity between the tutor and overall summaries is the highest. A possible reason for this could be that the tutor provides more detailed in-

---

[2]https://github.com/doccano/doccano

| Dialogue History | Generic Summary | Student Summary | Tutor Summary |
|---|---|---|---|
| T : *Please try to fill in the blank in Italian.* 
 S : *e di fronte al conigli viola.* 
 . 
 . 
 . 
 T : *Oh! You almost got it! Very good. You just left off the "o" on the word "coniglio." It makes it masculine and singular.* | *Tutor tells student to fill in the blank in Italian. Student answers "e di fronte al conigli viola".which is almost correct. The tutor says that the student just left off the "o" on the word "coniglio" which makes it masculine and singular.* | *Student answers "e di fronte al conigli viola".which is almost correct* | *Tutor tells student to fill in the blank in Italian. The tutor says that student has just left off the "o" on the word "coniglio" which makes it masculine and singular.* |

Figure 2: Sample dialogue history and its perspective based summaries from the *CIMA-Summ* dataset

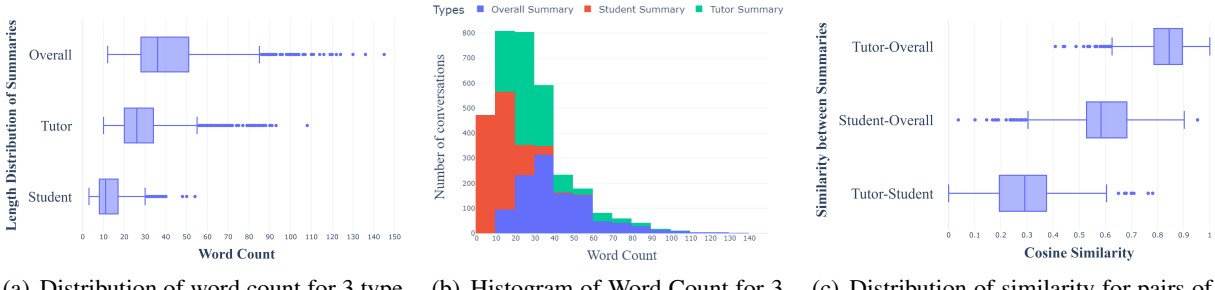

(a) Distribution of word count for 3 type of summaries

(b) Histogram of Word Count for 3 type of summaries

(c) Distribution of similarity for pairs of summaries

Figure 3: Distribution statistics of the *CIMA-Summ* dataset

formation containing a higher level of complexity. Also, tutor and student summaries are least similar, possibly because students concentrate only on the most salient ideas while taking notes. Figure 3(b) represents a histogram of word count for these three type of summaries. For 98% of the conversations, tutor's summary is longer than the student's summary, reflecting a deeper understanding & comprehensive analysis of context by the tutor.

## 4 Proposed Methodology

In this section, we discuss the problem statement and proposed methodology in detail.

**Problem Description.** Our proposed task *Multimodal Perspective based Dialogue Summarization (MM-PerSumm)* is defined as follows :

**Input :** The input comprises of three entries: (1) The source dialogue $D = \{T_0 < Ac_0 >, S_0 < In_0 >, T_1 < Ac_1 >, S_1 < In_1 >, ...\}$, which is a conversation between a VT and a student where $T_i$ and $Ac_i$ represents the VTs utterance and its corresponding action tag. $S_i$ and $In_i$ represents the student's utterance and its corresponding intent tag; (2) An associated visual image $I$, which is used by the tutor to facilitate the teaching process; (3) A specified perspective, $P$, which represents the viewpoint (either student, tutor, or a neutral observer) from which the dialogue is to be summarized.

**Output :** The output is a natural language sequence, $Y$, representing the summarized dialogue from the perspective, $P$. Given an instance of $D$, $I$, and $P$, $Y$ can be manifested as a diverse summary

conditioned on the perspective and image.

### 4.1 Image and Perspective-guided Dialogue Summarization (*IP-Summ*) Model

This section presents the novel architecture of **IP-Summ**, a perspective-guided, multimodal Seq2Seq model (Figure 4) that synthesizes dialogue and image context to enhance educational dialogue summarization. *IP-Summ* comprises of three distinct modules: (1) *Global Context Encoder*, (2) *Perspective Context Encoder*, and (3) *Contextual Fusion Module* discussed below.

#### 4.1.1 Global Context Encoder

This integral component of our model serves as the primary mechanism for understanding the overall essence and the key points of the dialogue. It processes the complete multi-turn dialogue, $D$ along with the corresponding image, $I$ and generates a global context representation, $Z_G$ providing a high-level understanding of the conversation.

**Dialogue Context Encoder.** Firstly, the source dialogue, $D$, is fed into a language encoder, $L_{enc}(\cdot)$. This language encoder is a function that maps the raw dialogue text into a high-dimensional space. The output of this operation is a vector representation, $H_D$ of the dialogue.

**Image Encoder.** In parallel, we also feed the associated image, $I$ into a vision encoder, $V_{enc}(\cdot)$. This encoder is a function that processes the image and converts it into a high-dimensional vector representation, $H_I$.

**Dimensionality Reduction.** Following their

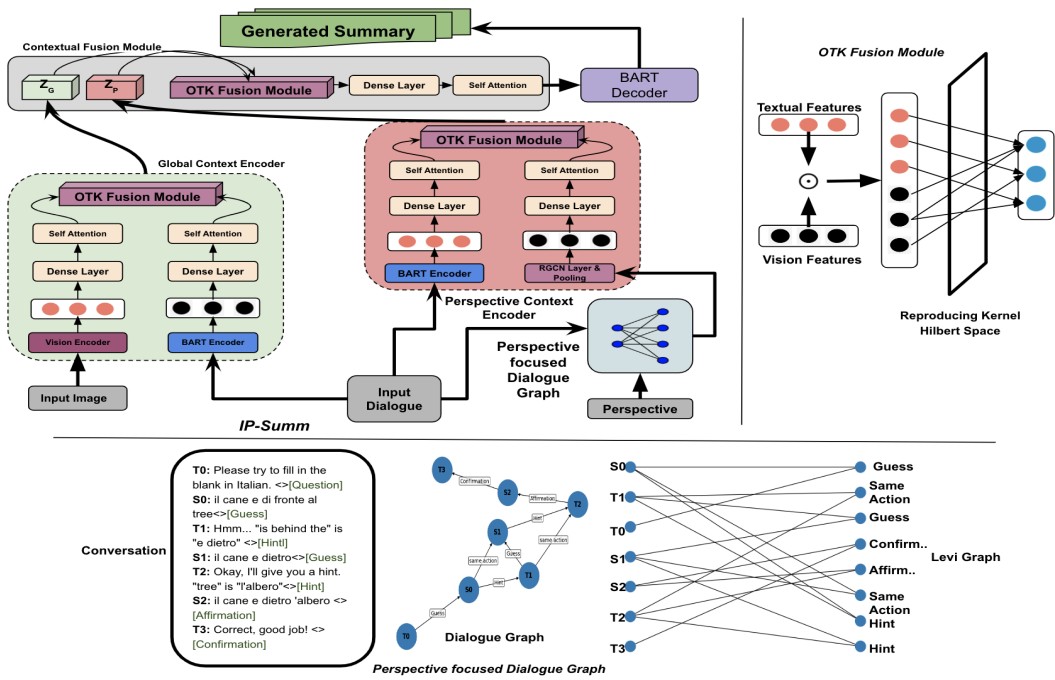

Figure 4: Overall architecture of *IP-Summ*

generation, these text and image representations undergo a dimensionality reduction procedure. This is facilitated through a nonlinear transformation, succeeded by a self-attention layer, as captured by the following equations:

$$H_D^{'} = SoftMax(\frac{H_D H_D^T}{\sqrt{d}})H_D \qquad (1)$$

$$H_I^{'} = SoftMax(\frac{H_I H_I^T}{\sqrt{d}})H_I \qquad (2)$$

**Multi-modal Fusion.** Following this, the two vectors are concatenated and passed through an Optimal Transport-based Kernel Embedding (OTKE) layer (Mialon et al., 2021). This layer aims to foster cross-modal interaction between the textual and visual contexts. The OTKE layer facilitates this by mapping the feature vectors into a Reproducing Kernel Hilbert Space (RKHS) (Berlinet and Thomas-Agnan, 2011), subsequently implementing a weighted pooling mechanism. This mechanism leverages weights that are determined by the transport plan between the set and a learnable reference. This operation results in a single vector representation, $Z_G$, which is given by:

$$Z_G = OTKE([H_D^{'} : H_I^{'}]) \qquad (3)$$

This vector, $Z_G$ encapsulates the global context, capturing both the textual and visual features of the dialogue-image set.

### 4.1.2 Perspective Context Encoder

Depending upon the specified perspective (i.e., student, tutor, or neutral observer), this encoder processes the dialogue and generates a perspective-specific context representation. This ensures that the nuances, intentions, and actions associated with the chosen perspective are accurately captured and represented.

**Perspective driven Dialogue Graph Construction.** The Dialogue Graph, denoted as $G = (V, E)$, comprise of nodes, $V$ representing utterances and edges, $E$ illustrating the transition between these utterances. The methodology is intrinsically adaptable to the specified perspective, $P$ (i.e., general, student-focused, or tutor-focused), allowing us to construct graphs tailored to different perspectives discussed as follows :

• **Node Creation :** Independent of the perspective, $P$, every utterance within the dialogue, $D$ is transformed into a node in $G$. Every node, $v \in V$ represents an utterance, $(S_i)$ or $(T_i)$, where $S_i$ and $T_i$ stand for student's and tutor's utterances, respectively.

• **Edge Creation :** The formulation of edges in graph, $G$ is intrinsically tied to the chosen perspective, $P$. Representing an edge from node $i$ to node $j$ as $e_{\{ij\}}$, each edge is attributed with a label, $l_{\{ij\}}$, reflective of the corresponding action or intent tag.

**i) General Perspective :** For an edge between each consecutive pair of utterances in a dialogue

of length $N$, we establish: $e_{\{ij\}}$ for all $i$, $j$ such that $j = i + 1$ and $i \in \{0, 1, ..., N - 2\}$. The label, $l_{\{ij\}}$ is assigned as per equation 4. Additional edges, $e_{\{xy\}}$, are interjected between nodes associated with identical action/intent tags.

**ii) Student-focused Perspective :** When our perspective, $P$ is focused on the student, the edge creation process remains invariant, but the label assignment function prioritizes the student's intent tags (Equation 4). We also introduce an additional edge, $e_{\{xy\}}$ labeled 'same intent' for every pair of student utterances $(x, y)$ that share the same intent.

**iii) Tutor-focused Perspective :** When $P$ is tutor-focused, our label assignment function prioritizes the tutor's action tags ((Equation 4)). Under this perspective, an additional edge, $e_{\{xy\}}$ labeled 'same action' is introduced for each pair of tutor utterances $(x, y)$ that share the same action tag.

$$l_{ij} = \begin{cases} Ac_i & \text{if utt. } i \in \text{T \& } P \in 'T','Gen' \\ In_i & \text{if utt. } i \in \text{S \& } P \in 'S','Gen' \\ 'context' & \text{if utt. } i \in \text{S \& } P =' T' \\ 'context' & \text{if utt. } i \in \text{T \& } P =' S' \end{cases}$$
(4)

where *context* label indicates that the utterance provides contextual information to the student's/tutor's responses.

**Levi Graph Generation.** Upon the completion of the directed graph, $G$ generation, the next step is to construct the corresponding Levi graph, $L(G)$ inspired from graph theory. A Levi graph is a representation that preserves the relational structure of the initial dialogue graph but further emphasizes the linkage between utterances sharing the same intent or action. Specifically, $L(G)$ is obtained by introducing additional vertices corresponding to the edges of $G$. Each edge, $e_{ij}$ in $G$ becomes a vertex, $v_{ij}$ in $L(G)$, where $i$ and $j$ are utterances in the dialogue. Consequently, if two edges, $e_{ij}$ and $e_{kl}$ in $G$ share a node ($j = k$), they are connected in $L(G)$ by an edge $e_{v_{ij}v_{kl}}$. Further, we preserve the context by adding edges between new vertices corresponding to consecutive turns in the dialogue. If $e_{ij}$ and $e_{jk}$ are consecutive turns in the dialogue in $G$, we add an edge, $e_{v_{ij}v_{jk}}$ in $L(G)$, hence maintaining the sequential structure of the conversation. This transformation from $G$ to $L(G)$ allows for a more nuanced view of the dialogue's dynamics, highlighting both the sequential and structural aspects of the conversation, as it captures not only the utterance transitions but also the shared intents or actions across different

turns. The Levi graph, $L(G)$, with nodes denoted by $V$, is initially embedded using word2vec, producing node embeddings of dimension $d$. This graph is then subjected to several rounds of convolutions via a relational graph convolutional network (Schlichtkrull et al., 2017). After three convolution rounds, a max-pooling graph operation is used to generate graph embedding, $H_G$. In parallel, a vector representation, $H_D$ of the dialogue is generated. Following this, both $H_D$ and $H_G$ are subjected to dimensionality reduction via a nonlinear transformation followed by a self-attention layer, resulting in $H_G'$ and $H_D'$. Lastly, the OTKE layer infuses the perspective graph into the language representations of the dialogue, yielding the final vector, $Z_P$. This vector encapsulates the dialogue's language representations along with perspective-specific nuances.

### 4.1.3 Contextual Fusion Module

The Contextual Fusion Module serves as an integrator, fusing the information from both the Global and Perspective Context Encoder. We utilize the OTKE fusion module to fuse the context representations, establishing a synergy between the global and perspective-specific contexts. The vectors, $Z_G$ and $Z_P$ from the Global and Perspective Context Encoder, respectively, undergoes the OTKE operation to create a fused representation, $Z_{Final}$, which ensures a more nuanced understanding of the dialogue by capturing and integrating diverse insights. The vector, $Z_{Final}$ is then subjected to a multi-headed self-attention mechanism to yield $Z_{Final}'$. This final vector, $Z_{Final}'$ is subsequently processed by a standard BART decoder, transforming the context-rich vector representation into a dialogue summary.

## 5 Experiments and Results

This section details the experimental setup followed by results and analysis of *IP-Summ*.

### 5.1 Experimental Setup

The training was conducted for 20 epochs with a learning rate of 5 x $10^{-5}$, a batch size of 16, using the Adam optimizer, and an Adam epsilon value of 1 x $10^{-8}$. Our proposed model, as well as all the ablated models, are built on top of the BART-Base (Lewis et al., 2019) architecture. We split the 1,000 articles from CIMA Dataset into 80% for training, 10% for validation, and 10% for testing. The performance of the generative models was evaluated using several metrics, including aver-

| | Model | Overall Summary | | | Student Summary | | | Tutor Summary | | |
|---|---|---|---|---|---|---|---|---|---|---|
| | | R | B | BS | R | B | BS | R | B | BS |
| **Baselines** | BART | 0.31 | 0.25 | 0.74 | 0.39 | 0.4 | 0.72 | 0.27 | 0.22 | 0.71 |
| | T5 | 0.2 | 0.23 | 0.72 | 0.25 | 0.33 | 0.71 | 0.22 | 0.21 | 0.72 |
| | DialoGPT | 0.25 | 0.21 | 0.77 | 0.36 | 0.4 | 0.72 | 0.27 | 0.23 | 0.75 |
| | MultimodalBART | 0.38 | 0.26 | 0.81 | 0.43 | 0.42 | 0.78 | 0.33 | 0.25 | 0.81 |
| | Con_Summ | 0.25 | 0.24 | 0.77 | 0.39 | 0.41 | 0.82 | 0.31 | 0.24 | 0.81 |
| **Ablation Model** | IP-Summ - ViT | 0.42 | 0.26 | 0.87 | 0.54 | 0.46 | 0.84 | 0.39 | 0.28 | 0.84 |
| | IP-Summ - RGCN | 0.43 | 0.28 | 0.87 | 0.56 | 0.45 | 0.88 | 0.41 | 0.28 | 0.85 |
| | IP-Summ - LGG | 0.42 | 0.26 | 0.85 | 0.55 | 0.45 | 0.87 | 0.4 | 0.27 | 0.85 |
| **Proposed Model** | **IP-Summ** | **0.46** | **0.3** | **0.91** | **0.59** | **0.49** | **0.93** | **0.43** | **0.31** | **0.9** |

Table 1: Comparison of *IP-Summ* with other baselines and ablation models on automated metrics. R: ROUGE-L score, B: Average BLEU Score, BS: BERTScore F1 and LGG: Levi Graph Generator

age BLEU score (Papineni et al., 2002), ROUGE-L score (Lin, 2004), and BERTScore F1 (Zhang et al., 2020a). Furthermore, we conduct a human evaluation of the most effective models. We evaluate the model's output on the following aspects: *Fluency*, *Informativeness*, and *Relevance*. To assess these models in terms of human evaluation, three independent human users from the authors' affiliation were asked to rate 100 simulated dialogues on a scale of 1 (worst), 3 (moderate), and 5 (best) based on the above-mentioned criteria. The final reported score is the average of the human-rated scores. In our study, we used the following models as baselines: (1) BART model (Lewis et al., 2019), (2) T5 model (Raffel et al., 2020), (3) DialoGPT model (Zhang et al., 2020b), (4) MultimodalBART (Yu et al., 2021), and (5) Con_Summ (Liu and Chen, 2021), which is the only work to date on perspective-based conditional dialogue summarization. Please refer to Appendix A.3 for details.

## 5.2 Results and Discussion

A careful review of Table 1 reveals that our proposed model, *IP-Summ*, excelled across all metrics and perspectives. This model embodies excellent capabilities in text summarization, as demonstrated by its impressive scores: a Rouge score of 0.46, a BLEU score of 0.3, and a BERTScore of 0.91 when assessed from a general viewpoint. *IP-Summ*'s strong performance remains consistent when evaluated from the perspectives of both the student and the tutor. This implies the model's adaptability in understanding and summarizing various perspectives, a feature that can be credited to the utilization of the global and perspective context encoder.

**Comparison with the Baselines.** Delving into the baseline models, MultimodalBART emerged as the front-runner among this group. It scored the highest across all evaluation metrics - Rouge (0.38), BLEU (0.26), and BERTScore (0.81) - presenting a respectable benchmark for comparison. Among the

baseline models, BART and DialoGPT delivered relatively comparable results across BLEU and BERT-F1 scores, though BART displayed a slight edge in Rouge scoring. Conversely, T5 lagged in performance across all metrics.

With the help of ablation, we obtained crucial insights into the role of individual components within the *IP-Summ* model. When the ViT component was omitted, there was a significant reduction in the metric scores across all perspectives. This highlights the effectiveness of our proposed global context encoder, which is instrumental in integrating visual context into the dialogue. We observed a similar trend when the RGCN and Levi Graph Generator components were excluded from *IP-Summ*. Without the RGCN, there was a consistent decrease across all metrics and summaries, which underscores the importance of the dialogue graph encoder. The same trend was evident when we substituted the Levi graph with a standard graph, indicating that the Levi graph encapsulates richer information about the perspective compared to a standard dialogue graph. Additionally, a pattern emerges when we examine the scores across the student and tutor perspectives. For all models evaluated, including *IP-Summ*, the scores for the student perspective are consistently higher than those for the tutor perspective. This pattern is indicative of the relative complexity of generating tutor summaries compared to that of student summaries.

**Effect of different Components.** In addition, we have illustrated the impact of various vision encoders, fusion mechanisms, and graph encoders on our model in Figure 5. (1) When implemented with the Vision Transformer (ViT) (Dosovitskiy et al., 2021), our model demonstrated superior performance across all perspectives compared to other vision models. This highlights the efficacy of transformer-based vision models in effectively modeling image features. The subsequent most

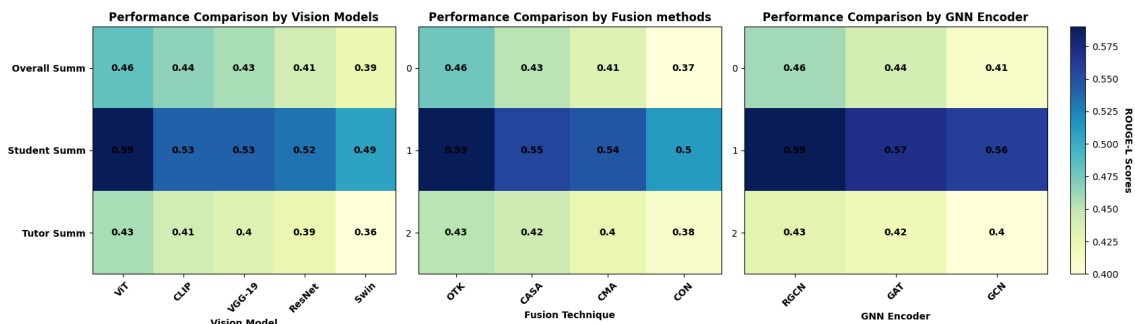

Figure 5: Heatmap displaying variations in ROUGE scores and illustrating the impact of altering different components within the proposed model. (CASA:Context-Aware Self Attention, CMA: Cross-Modal Attention, CON: Concatenation)

| Model | Overall Summary | | | Student Summary | | | Tutor Summary | | |
|---|---|---|---|---|---|---|---|---|---|
| | FL | IN | RE | FL | IN | RE | FL | IN | RE |
| BART | 4.0 | 3.6 | 3.3 | 4.2 | 4.1 | 3.3 | 3.9 | 3.2 | 3.4 |
| MultimodalBART | 4.2 | 4.3 | 3.9 | 4.5 | 4.5 | 3.8 | 3.9 | 4.0 | 3.9 |
| IP-Summ | 4.3 | 4.5 | 4.3 | 4.5 | 4.6 | 4.2 | 4.1 | 4.2 | 4.4 |

Table 2: Human evaluation scores of proposed model, *IP-Summ* and corresponding best baselines. FL:Fluency, IN:Informativeness, RE:Relevance

successful vision model was CLIP (Radford et al., 2021), trailed by VGG-19 (Simonyan and Zisserman, 2015), ResNet (He et al., 2015), and Swin (Liu et al., 2021) vision models. This sequence indicates the dominance of ViT and CLIP over more conventional image encoders. (2) The incorporation of the OTK fusion module into our model resulted in a consistently higher performance across all metrics compared to other fusion techniques, thus, illustrating the superiority of the OTK fusion mechanism. Following the OTK fusion module, the next most effective mechanism was context-aware self-attention (Yang et al., 2019), which was then succeeded by cross-modal attention and, finally, concatenation. This pattern demonstrates how the utilization of a more complex, sophisticated, and robust fusion mechanism can significantly enhance the results. (3) The use of the RGCN encoder led to improved performance in comparison to other graph encoders, such as GAT (Veličković et al., 2018) and GCN (Kipf and Welling, 2017). This superior performance may be attributed to RGCN's ability to handle multi-relation graphs, making it an ideal choice for our Levi graph which contains multiple relations and nodes, unlike a standard dialogue graph.

**Human Evaluation.** Table 2 presents the results for BART, MultimodalBART, and *IP-Summ*. In terms of fluency, all models demonstrated admirable performance. *IP-Summ* led this category with a score of 4.3, followed closely by MultimodalBART (4.2) and BART (4). The metric's high scores across all models underscore their ef-

fectiveness in language generation. However, the picture becomes more differentiated when considering the metric of informativeness. Here, the BART model, scoring 3.6, did not perform as well as its counterparts. The absence of a visual context component in BART is likely to be a factor, as this component plays a crucial role in generating summaries rich in key information. By contrast, the *IP-Summ* and MultimodalBART models, incorporating a visual context component, displayed enhanced performance in this area, achieving scores of 4.5 and 4.3, respectively. As for the metric of relevance, the *IP-Summ* model showed superior performance with a score of 4.3. The model's perspective encoder, which aligns the summary with the specific perspective, likely plays a significant role in this outcome. The scores achieved by MultimodalBART (3.9) and BART (3.3) in this area underline the importance of incorporating perspective-specific encoding mechanisms in models tasked with generating relevant summaries. To ensure the reliability of our human and automatic evaluations, we computed inter-annotator agreement for human judgments and statistical significance tests for metric improvements. For human evaluation, we calculated the rate of majority agreement across annotators on the acceptability of model responses. This yielded an inter-annotator agreement score of 72.3%, indicating reliable consensus. For automatic metrics, we conducted the statistical significance Welch's t-test at 5% significance to ensure that the improvement of our proposed model over the baselines is reliable. All the reported results are statistically significant.

**Qualitative Analysis of Generated Summaries.** In Figure 6, we showcase two instances of our model-generated student, tutor, and generic summaries, alongside the ground truth summaries. The results demonstrate that the generic summaries generated by our model, *IP-Summ*, align well with the

| Dialogue History | Ground Truth Generic Summary | IP-Summ Generic Summary | Ground Truth Student Summary | IP-Summ Student Summary | Ground Truth Tutor Summary | IP-Summ Tutor Summary |
|---|---|---|---|---|---|---|
| **T:** Please try to fill in the blank in Italian **S:** how do you say behind? **T:** Okay, I\'ll give you a hint. "is behind the" is "e dietro la" **S:** and how do you say dog? **T:** OK, "box" is "scatola" **S:** How do you say dog? **T:** Dog is "il cane" | Tutor instructs to fill in the blank in Italian. Student asks Italian word for "behind" to which tutor gives hint that "is behind the" is "e dietro la". Student asks Italian words for "dog". Tutor says "box" is "scatola".Student repeats that what would be the Italian word for "dog" to which tutor replies "il cane" | Tutor asks student to fill in the blank in Italian. "Is behind the" is "e dietro la". Tutor explains student how to say box and scatola. Dog is called "il cane" by Tutor. | Student wants to know how to say for "behind" and "dog". | Student wants to know how to say behind and dog. Student asks teacher how to say dog. Teacher gives him examples. | Tutor instructs to fill in the blank in Italian.Tutor hints "is behind the" is "e dietro la". And tutor says in Italian "box" and "dog" are "scatola" and "il cane", respectively. | "Is behind the" is "e dietro la". Tutor: Box is "scatola" and Dog is "il cane". |
| **T:** Please try to fill in the blank in Italian. **S:** what is is behind? **T:** Well, "is behind the" is "e dietro la **S:** what is box? **T:** Well, "box" is "scatola **S:** what is yellow? **T:** Okay, I\'ll give you a hint. "yellow" is "gialla **S:** Is it e dietro la scatola giala? **T:** That\'s right, but you have a typo. gialla | Tutor instructs student to fill in the blank in Italian. Student asks Italian words for "is behind the", "box" and "yellow" to which tutor replies "e dietro la", "scatola" and "gialla", respectively. Student answers e dietro la scatola giala to which tutor suggests to correct the spelling of "gialla". | Tutor asks student to fill in the blank in Italian. "Is behind the" is "e dietro la" in Romanian. Tutor explains to student that "box" is "scatola". "Yellow" is "gialla" in Italian. "Giovanni" is not a correct translation. | Student asks Italian words for "is behind the", "box" and "yellow". Student answers "e dietro la scatola giala". | Student asks what is behind and what is in the box. | Tutor instructs student to fill in the blank in Italian. Tutor explains Italian words for "is behind the", "box" and "yellow" are "e dietro la", "scatola" and "gialla", respectively. After getting answers tutor suggests student to correct the spelling of "gialla". | "Is behind the" is "e dietro la". Tutor explains that box is "scatola" and yellow is "gialla". "Giovanni" is not a correct translation. |

Figure 6: Sample instance presenting the gold standard summaries and the summaries generated by *IP-Summ* from multiple perspectives

| Dialogue History | Ground Truth Generic Summary | BART Generic Summary | Ground Truth Student Summary | BART Student Summary | Ground Truth Tutor Summary | BART Tutor Summary |
|---|---|---|---|---|---|---|
| **T:** Please try to fill in the blank in Italian **S:** how do you say behind? **T:** Okay, I\'ll give you a hint. "is behind the" is "e dietro la" **S:** and how do you say dog? **T:** OK, "box" is "scatola" **S:** How do you say dog? **T:** Dog is "il cane" | Tutor instructs to fill in the blank in Italian. Student asks Italian word for "behind" to which tutor gives hint that "is behind the" is "e dietro la". Student asks Italian words for "dog". Tutor says "box" is "scatola".Student repeats that what would be the Italian word for "dog" to which tutor replies "il cane" | E dietro la" is how you say "is behind the" in Italian. the tutor is saying that "il cane" means dog | Student wants to know Italian words for "behind" and "dog". | A kid asks an instructor to teach him how to say 'dog' , 'tree' and 'behind', | Tutor instructs to fill in the blank in Italian.Tutor hints "is behind the" is "e dietro la". And tutor says in Italian "box" and "dog" are "scatola" and "il cane", respectively. | Tutor is teaching a foreign language to student with focus on dog. |
| **T:** Please try to fill in the blank in Italian. **S:** what is is behind? **T:** Well, "is behind the" is "e dietro la **S:** what is box? **T:** Well, "box" is "scatola **S:** what is yellow? **T:** Okay, I\'ll give you a hint. "yellow" is "gialla **S:** Is it e dietro la scatola giala? **T:** That\'s right, but you have a typo. gialla | Tutor instructs student to fill in the blank in Italian. Student asks Italian words for "is behind the", "box" and "yellow" to which tutor replies "e dietro la", "scatola" and "gialla", respectively. Student answers e dietro la scatola giala to which tutor suggests to correct the spelling of "gialla". | The tutor instructs the student in Italian, explaining that "is behind the" in Romanian is "e dietro la", "box" is "scatola, and "yellow" is "gialla". | Student asks Italian words for "is behind the", "box" and "yellow". Student answers "e dietro la scatola giala". | Student ask what is yello and ask for hints | Tutor instructs student to fill in the blank in Italian. Tutor explains Italian words for "is behind the", "box" and "yellow" are "e dietro la", "scatola" and "gialla", respectively. After getting answers tutor suggests student to correct the spelling of "gialla". | The tutor teaches that in Italian, "box" translates to "scatola", "yellow" to "gialla", and clarifies that "Giovanni" isn't a correct translation. |

Figure 7: Sample instance presenting the gold standard summaries and the summaries generated by the baseline *BART* model from multiple perspectives

ground truth summaries in both the given examples. However, there are instances where *IP-Summ* falters in other perspectives. For instance, in the first example, *IP-Summ*'s student summary mistakenly incorporates information from the tutor about "giving student examples." Additionally, we observe that *IP-Summ*'s tutor summary covers all the core concepts but lacks fluency compared to the student summary, indicating the inherent difficulty in generating tutor summaries as opposed to student summaries. Figure 7 shows sample generated summaries (for the same instances in that of figure 6) from the baseline BART model. *Some more analyses of the generated summaries are reported in the Appendix section*.

## 6  Conclusion

The rapid adoption of VTs has brought forth novel challenges and opportunities in enhancing the learning experience amongst which dialogue summarization stands out as a pivotal area warranting dedicated attention. Our work addresses this crucial aspect by proposing a new task of *Multi-modal Perspective-based Dialogue Summarization (MM-PerSumm)* in an educational setting, which paves the way for a holistic understanding of the VT-student exchanges. We introduce a novel dataset, *CIMA-Summ*, that features dialogue summaries from diverse perspectives, offering a multi-faceted view of the learning interaction. We further propose a model, *Image and Perspective-guided Dialogue Summarization (IP-Summ)*, which effectively incorporates both image context and dialogue perspectives into the summarization process.

## 7  Acknowledgement

Dr. Sriparna Saha gratefully acknowledges the Young Faculty Research Fellowship (YFRF) Award, supported by Visvesvaraya Ph.D. Scheme for Electronics and IT, Ministry of Electronics and Information Technology (MeitY), Government of India, being implemented by Digital India Corporation (formerly Media Lab Asia) for carrying out this research.

## Limitations

Despite its contributions, the present study acknowledges several limitations. The original CIMA dataset only includes 1134 conversations, which restricts us to generating summaries solely for these dialogues. This underlines the urgent need for more extensive and comprehensive datasets in a variety of educational settings. Furthermore, the specific application of our approach to language learning may not imply similar efficacy in other educational contexts, such as mathematical or science based learning. Therefore, further validation in these areas is essential to confirm the wider applicability of our proposed method. In addition, this study did not incorporate the potential use of Large Language Models (LLMs), which could offer additional insights and improvement of educational strategies. Future research should aim to overcome these limitations and further explore the potential of our approach.

## Ethical Considerations

In the course of this research, we committed to the highest standards of ethical conduct. The data used for this research was derived from a pre-existing, anonymized, and publicly available dataset, ensuring the privacy and confidentiality of the individuals involved. In the creation of our unique dataset, *CIMA-Summ*, we employed annotators who were compensated in accordance with institutional norms. We took great care to ensure that their work was scheduled within normal working hours, thus, promoting a healthy work-life balance. Moreover, the dataset was constructed in a way to ensure it does not promote or favour any demographic, race, or gender, thus, striving to mitigate the risk of potential bias in language models trained over this. The design and implementation of our model were guided by a robust commitment to avoiding potential harm. The *IP-Summ* model is purely a tool for dialogue summarization, and it does not involve any predictive or prescriptive functionalities that could impact individual users negatively or unfairly. This model also doesn't require to store any personal information about students and tutors, thus, preserving the privacy of users.

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

## A Appendix

### A.1 Frequently Asked Questions (FAQs)

✱ **What was the reason for utilizing the BART-base model as the underlying language model for IP-Summ?**

⇒ We selected BART-Base as the underlying model for our proposed approach based on comprehensive experimentation. Our findings consistently demonstrated that BART-Base outperforms T5-Base across various metrics and perspectives. While both BART-Base and DialoGPT-Base exhibited comparable performance, we opted for BART-Base due to its encoder-decoder architecture, which enables the encoding of diverse information prior to generating summaries. Moreover, in order to accommodate our computing resources, we specifically focused on the Base versions of all models for our comparative analysis.

✱ **What motivated us to utilize the CIMA dataset as the foundation dataset and expand it to incorporate summaries for dialogues?**

⇒ We specifically employed the CIMA dataset to construct the dataset for our proposed task, *MM-PerSumm*, for several compelling reasons. In comparison to other educational dialogue datasets (Caines et al., 2022), the CIMA dataset offers a wealth of contextual information pertaining to dialogues. Notably, it includes valuable features such as images, external knowledge, action tags, and intent tags. This comprehensive and multimodal nature of the CIMA dataset makes it exceptionally well-suited for developing a multimodal dialogue summarization dataset, thereby presenting a more relevant and impactful problem statement.

### A.2 Annotation Details

**Annotation Guidelines.** Annotators were guided to ensure that the created summaries were concise, yet informative, capturing the essence of the dialogue without excessive detail avoiding any personal interpretation or embellishment. Th detailed

Table 3: Annotation guidelines of *CIMA-Summ* dataset.

| S.No. | Annotation Guidelines |
|---|---|
| 1 | Review the provided image to understand the context of the dialogue. |
| 2 | Keep the intent-action labels in mind when writing the summaries as they provide valuable context to the dialogue. |
| 3 | For student summaries, focus on the key knowledge points, gained, misconceptions corrected, and progress in understanding. |
| 4 | For tutor summaries, note the teaching strategies used, the clarification of student doubts, and the overall guidance provided. |
| 5 | For overall summaries, condense the interaction, highlighting key dialogic exchanges, instructional elements, and learning outcomes. |
| 6 | Avoid incorporating personal opinions in any of the summaries. |
| 7 | Ensure summaries are concise, informative, and accurately reflect the dialogue content. |
| 8 | Summaries should not include personal interpretation or embellishment. |

approach ensures that the *CIMA-Summ* dataset offers rich, multi-perspective summaries that can serve as valuable resources for dialogue summarization in the education domain. A comprehensive summary of the annotation guidelines is presented in Table 3.

**Annotation Quality Assessment.** We assessed the quality of the summaries generated by the annotators following the approach described in Ghosh et al. (2022), which utilized two primary metrics : **Fluency :** It focuses on the linguistic correctness of the summaries using a 5-point Likert scale where a score of '1' indicated a summary riddled with grammatical errors while a score of '5' denoted a well-constructed summary free of grammatical mistakes.
**Adequacy :** This measures the content of the summary using a 5-point Likert scale. A score of '1' suggested that the summary either misrepresented or missed the intent-action labels along with the dialogue's main points. A score of '5' was given to a summary that accurately captured and reflected the salient points of the dialogue and the intent-action labels.

**Training of Annotators.** Given the specialized nature of dialogue summarization in an educational context, it was critical to train our annotators thoroughly to ensure the quality and consistency of the dataset. The annotators needed to understand not just the dialogue and its nuances but also the unique perspectives of the student and the tutor. Hence, a rigorous annotation training was undertaken. We employed a four-phase training process to ensure the proficiency and competence of the annotators. To initiate the annotation training, a set of dialogues from the *CIMA-Summ* dataset was pre-annotated to provide gold-standard samples. These examples were annotated with three type of summaries (student perspective, tutor perspective, and overall dialogue summary) by two experienced researchers

(from author's collaboration) specializing in educational dialogue systems and summarization. During each phase of training, the annotators were tasked with summarizing a selection of dialogues. They were reminded to follow the guidelines (Table 3) while generating the summaries. Post every phase, an evaluation of the summaries' fluency and adequacy was conducted. Feedback sessions were arranged after each phase where the annotators interacted with the experienced researchers to discuss the evaluations and ways to enhance the quality of the summaries. These iterative feedback discussions facilitated the development of their annotation skills, gradually improving the quality of the summaries. The annotation guidelines were also refined and updated after each phase, incorporating the insights obtained from these sessions. As a result of these iterative training phases, the quality of the summaries improved significantly from the first phase (fluency = 3.97, adequacy = 2.59) to the fourth phase (fluency = 4.91, adequacy = 4.81), indicating the effectiveness of our annotation training procedure.

**Annotation Process.** Each annotator was equipped with a secure account, permitting individual annotation and tracking of progress in the open-source platform Doccano[3]. Annotating dialogue summaries is a time-intensive task, requiring a balance between quality and efficiency. On average, our annotators needed 6-8 minutes to adequately summarize each dialogue instance, encapsulating necessary details and understanding the intent-action labels. Given the intricacy of the task, an honorarium of 10 INR per summarized dialogue instance was provided. The original dataset comprises 1134 dialogue instances, all of which were incorporated into the annotation process. We followed a structured schedule over a span of five days in a week as :

**Day 1 and Day 4:** Each annotator was assigned 30 dialogues for summarization, split into three batches of 10 summaries, with a mandatory 20-minute break between each batch to ensure focused and efficient work.

**Day 2 and Day 5:** Annotators were tasked with evaluating summaries produced by their peers, based on the quality assessment criteria outlined above.

**Day 3:** Feedback sessions were held with the annotators to discuss potential improvements

---

[3]https://github.com/doccano/doccano

and address any challenges they faced during the annotation process.

### A.3 Baseline Models

In this section, we will discuss the training process for each baseline model used in our study.

- All three of these models — BART, T5, and DialoGPT — are unimodal, meaning they are trained to handle one modality of data, in this case, text. For the purpose of our study:

  - Tutor Perspective Training: The training dataset was structured such that the input consists of a conversation, and the output is the corresponding tutor summary.
  - Student Perspective Training: Following a similar structure, the input is the conversation, and the output is the student's summary.
  - Overall Perspective Training: For a holistic perspective, the training data once again had conversations as input with the overall summary as the output.

  These models were then trained using the Maximum Likelihood Estimation (MLE) objective function.

- Multimodal BART: The Multimodal BART extends the original BART model by incorporating both text and image modalities. The training process for each perspective is detailed below:

  - Tutor Perspective Training: The input is a combination of conversation text paired with an image, and the output is the corresponding tutor summary.
  - Student Perspective Training: Similar to the tutor perspective, the input pairs the conversation with an image, and the output is the student's summary.
  - Overall Perspective Training: The holistic view takes the conversation and image as input and provides the overall summary as output.

  An intrinsic feature of this model is the cross-modal attention mechanism within the encoder. This mechanism fuses the representations of text and images, which is subsequently fed into the decoder for further processing. The training objective remains the Maximum Likelihood Estimation (MLE).

| Dialogue History | Generic Summary | Student Summary | Tutor Summary |
|---|---|---|---|
| **T:** *Pink is rosa. Fill in the blank in Italian.* **S:** *What is tree & behind in Italian again?* **T:** *Tree is l'albero & behind the is e dietro. If not, what's the correct answer?* **S:** *il cane e dietro rosa l'albero?* **T:** *I (the) is prepended to the following word when it begins with a vowel.* **T:** *Look at your order of words again. Adjectives (such as color words) follow the noun they modify in Italian.* | *The tutor instructs the student to fill in the blank in Italian and dictates that Pink is rosa. The student asks the tutor about the Italian word for tree and behind which the tutor replies as l'albero and dietro respectively. The student's answer is il cane dietro rosa l'albero? to which the tutor replies that l' is prepended to the following word when it starts with a vowel. The student asks for the correct answer to which the tutor suggests to look at the word order again* | *The student asks the tutor about the Italian words for tree and behind. The student responds with il cane e dietro rosa l'albero? which is incorrect. Then the student asks for the correct answer* | *The tutor dictates that Pink is rosa in Italian. Tree and is behind the are l'albero and e dietro respectively. The tutor replies that I is prepended to the following word when it begins with a vowel. Furthermore, the tutor suggests the student to look at the order of the words and tells him that adjectives follow the noun in Italian.* |

Figure 8: Example dialogue history and its corresponding summaries from the *CIMA-Summ* dataset

- Con_Summ: It employs a single BART model adapted for the task of generating perspective-controlled summaries from input dialogues. The model requires a two-entry input: Source Input Dialogue: This is the primary conversation that needs to be summarized. Perspective Token: This token indicates the desired perspective (tutor, student, or overall) for which the summary is to be generated. Given these inputs, the output is a summary corresponding to the designated perspective token.

### A.4 *CIMA-Summ* Dataset

Figure 8 illustrates another example instance from our proposed *CIMA-Summ* dataset.

### A.5 Experimental Section

The models were trained on a Tyrone machine equipped with an Intel Xeon W-2155 processor and an 11 GB Nvidia 1080Ti GPU. All the models were implemented using Scikit-Learn and PyTorch.

### A.6 Analysis and Discussion

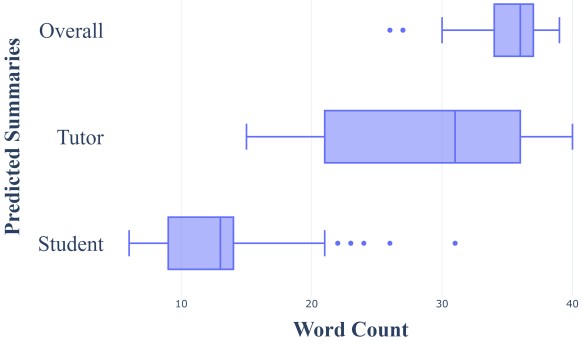

Figure 9: Distribution of Length for 3 types of Predicted Summaries

**Linguistic Analysis of Generated Summaries.**
Figure 9 presents the distribution of length for 3 types of generated summaries. The generated summaries have relatively shorter length compared to

the ground truth of the complete dataset as shown in Figure 3(a). The generated summaries have a more condensed length distribution compared to the ground truth of the complete dataset i.e. generated summaries show less variability in summary lengths across all three categories (Student, Tutor, and Overall). In Figure 10, we can notice an overall trend where the number of English and corresponding Italian word occurrences are highest in the Overall predicted summary followed by Tutor and Student generated summaries. A similar trend was noticed in the word count distribution for the annotated *CIMA-Summ* dataset. Also, there are no occurrences of few words in Tutor's summaries like *'green'* & it's Italian translation *'verde'*, *'bed'* & it's Italian translation *'letto'*, etc. While they do appear in student summaries indicating that the student's were able to figure out these Italian words without tutor's aid and tutor only focused on improving areas were student needed help and correction.

**Qualitative Analysis of Generated Summaries.**
In Figure 6, we showcase two instances of our model-generated student, tutor, and generic summaries, alongside the ground truth summaries. The results demonstrate that the generic summaries generated by our model, *IP-Summ*, align well with the ground truth summaries in both the given examples. However, there are instances where *IP-Summ* falters in other perspectives. For instance, in the first example, *IP-Summ*'s student summary mistakenly incorporates information from the tutor about "giving student examples." Additionally, we observe that *IP-Summ*'s tutor summary covers all the core concepts but lacks fluency compared to the student summary, indicating the inherent difficulty in generating tutor summaries as opposed to student summaries. Figure 7 shows sample generated summaries (for the same instances in that of figure 6) from the baseline BART model.

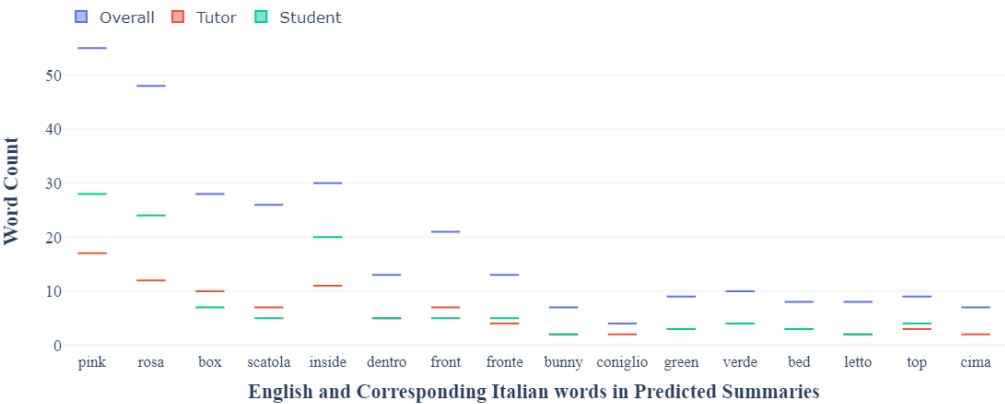

Figure 10: Frequency of top English and corresponding Italian words in predicted summaries