# OpenReview forum: "Can you Summarize my learnings? Towards Perspective-based Educational Dialogue Summarization"
_EMNLP/2023/Conference — EMNLP 2023 Findings_

### Official Review · Reviewer_q3T3 · 2023-08-04

**Soundness:** 4

**Excitement:**

2: Mediocre: This paper makes marginal contributions (vs non-contemporaneous work), so I would rather not see it in the conference.

**Paper Topic And Main Contributions:**

The paper discusses the increasing use of Virtual Tutors (VT) in recent years, which has enabled more efficient, personalized, and interactive AI-based learning experiences. A crucial aspect of educational chatbots is summarizing the conversations between VT and students, as it helps consolidate learning points and monitor progress. The paper proposes a new task called Multi-modal Perspective-based Dialogue Summarization (MM-PerSumm) in an educational setting. They introduce a novel dataset, CIMA-Summ, which contains educational dialogues summarized from three unique perspectives: Student, Tutor, and Generic.

To address the MM-PerSumm task, the paper presents the Image and Perspective-guided Dialogue Summarization (IP-Summ) model. This model is a SeqSeq language model that incorporates multi-modal learning from images and a perspective-based encoder. The perspective-based encoder constructs a dialogue graph, capturing the intentions and actions of both the VT and the student. By leveraging both textual and visual information, the IP-Summ model allows for more comprehensive and accurate summarization of educational dialogues from diverse perspectives.

The paper emphasizes the importance of summarizing dialogues from multiple perspectives to gain valuable insights for improving AI-based educational systems and enhancing students' learning outcomes. Summaries can provide essential feedback for the VT's teaching efficiency and suggest areas of improvement. For learners, summaries distill key learning points, track their progress, and inform their future study paths. The proposed MM-PerSumm task and IP-Summ model hold significant potential in advancing AI-driven education by revealing diverse insights and contributing to the evolution of virtual tutoring systems.

**Reasons To Accept:**

The paper introduces a new and innovative task of MM-PerSumm in an educational setting. This task involves summarizing educational dialogues from three unique perspectives: the Student, the Tutor, and a Generic viewpoint. By considering different viewpoints, the summarization process can provide diverse insights into the virtual tutoring interactions and contribute to the improvement of AI-based educational systems.

The paper extends an existing educational conversational dataset, CIMA, by summarizing the dialogues from three different perspectives (Student, Tutor, and Generic viewpoint) to create CIMA-Summ.

Image and Perspective-guided Dialogue Summarization (IP-Summ) Model: The paper proposes the IP-Summ model, which is a SeqSeq language model incorporating multi-modal learning from images and a perspective-based encoder.


**Reasons To Reject:**

1）Can the author provide some case studies in experiments to better demonstrate the effect of the model.
2）I didn't particularly feel the benefits of summarizing the action content of different characters in the educational dialogue. Instead, I felt that the data became discontinuous and I couldn't see the logical relationship of the summary content. Should the other party's behavior be weakened in it, but not completely deleted?

**Reproducibility:**

4: Could mostly reproduce the results, but there may be some variation because of sample variance or minor variations in their interpretation of the protocol or method.

**Reviewer Confidence:**

3: Pretty sure, but there's a chance I missed something. Although I have a good feel for this area in general, I did not carefully check the paper's details, e.g., the math, experimental design, or novelty.

---

> ### Author Rebuttal · Authors · 2023-08-29
>
> We sincerely thank the reviewer for their valuable review.
>
>
> **Weakness 1: Can the author provide some case studies in experiments to better demonstrate the effect of the model.**
>
> Response 1:  We do already present a comprehensive qualitative analysis in Appendix A.5 of the paper, including: Figure 9 showing generated summaries from our model vs gold summaries for a sample instance, across all perspectives. This illustrates our model's ability to produce varied, nuanced summaries capturing key details from different viewpoints. Figure 10 contrasting our model's outputs against a baseline model, for the same instance. A few key trends demonstrate IP-Summ's advantages:
>
> * IP-Summ produces more informative and relevant generic summaries. This is likely due to its use of multimodal fusion to incorporate visual information and perspective dialogue graphs to maintain conversational flow.
> * BART summaries tend to be vague and miss key details that IP-Summ captures through its richer encoding mechanisms.
> * IP-Summ better targets perspective-relevant information through its graph-based modeling. BART lacks this perspective grounding.
> * IP-Summ summaries are more coherent and complete in covering all salient information. BART shows fragmentation across turns.
>
>  In summary, the case study analysis highlights IP-Summ's ability to generate improved summaries by leveraging both visual and conversational context through dedicated modeling components.
>
> Let us know if you would like us to include any additional details or case examples to further demonstrate the qualitative improvements from our approach. We are happy to expand this analysis in the paper based on your feedback.
>
> **Weakness 2: I didn't particularly feel the benefits of summarizing the action content of different characters in the educational dialogue. Instead, I felt that the data became discontinuous and I couldn't see the logical relationship of the summary content. Should the other party's behavior be weakened in it, but not completely deleted?**
>
> Response 2: Thank you for the thoughtful feedback. You raise a fair point - when summarizing from different perspectives, there is a risk of losing continuity if the content is fragmented too much across roles. However, I want to reassure you that our approach does maintain a complete, logical flow in the generated summaries. The key here is the Perspective Context Encoder module, which uses a perspective-focused graph to model relationships between all dialogue turns. This helps preserve continuity even when summarizing from particular viewpoints. To clarify further - we do not completely delete or ignore the other party's contributions when generating role-specific summaries. Rather, the graph encoding retains the full context to keep the logical flow intact. The perspective simply changes which details are focused on more in the summary.

---

### Official Review · Reviewer_gdPt · 2023-08-05

**Soundness:** 3

**Excitement:**

4: Strong: This paper deepens the understanding of some phenomenon or lowers the barriers to an existing research direction.

**Paper Topic And Main Contributions:**

This paper proposes the new task of Multi-Modal Perspective-based Educational Dialogue Summarization.  The task involves summarizing a student-virtual tutor dialogue with 3 different perspectives: student, tutor, and generic.  Summarizing from the student’s perspective focuses on the student’s progress and outcomes.  From the tutor’s perspective focuses on the tutoring strategy and efficacy.  The authors propose a model to approach this task and analyze performance.


**Questions For The Authors:**

* Did you consider conducting similarity among pairs of summaries in the same category?  Since some tutoring utterances in CIMA dialogues were constructed using the VT, I wonder if this would result in similar summaries.  I would also be curious in these results for all categories to ground the results presented in 3C.
* As CIMA contains both an image and related vocabulary and grammar information, did you consider whether to represent the student’s problem using the image or text representation?
* It’s unclear to me how the baseline models were trained for the task.  Could you please provide these details?  Did you fine tune these models on the task?  This can be added into the appendix, but is necessary information
* Did you conduct any inter-annotator agreement or significance testing for results reported in Table 2?


**Reasons To Accept:**

* Authors propose an important, novel task which aims to summarize Student-Virtual tutor conversations
* Authors augment an existing dataset with multiple summaries from different perspectives
* Authors propose interesting model to approach this task, which includes a perspective-based dialogue graph taking dialogue actions into account.  Authors ablate the different components of their approach and produce a comprehensive analysis.

**Reasons To Reject:**

* While I love the premise of this task, I wish the authors had taken the tutor summaries a step further.  Instead of stating factually what the tutor has done (as shown in examples), why not include summaries of the educational techniques being deployed?  Since the introduction motivates tutor summaries being used to examine VT effectiveness, I feel the dataset falls short of this.


**Reproducibility:**

2: Would be hard pressed to reproduce the results. The contribution depends on data that are simply not available outside the author's institution or consortium; not enough details are provided.

**Reviewer Confidence:**

4: Quite sure. I tried to check the important points carefully. It's unlikely, though conceivable, that I missed something that should affect my ratings.

**Typos Grammar Style And Presentation Improvements:**

* How were fluency and adequacy ratings measured (line 235-238)?  Even if this is included in the appendix, it is good to include a minimal explanation if you are including the result in the main paper
* 933: Typo: the
* 400: backwards ‘

---

> ### Author Rebuttal · Authors · 2023-08-29
>
> We sincerely thank the reviewer for their valuable review.
>
>
> **Weakness 1: While I love the premise of this task, I wish the authors had taken the tutor summaries a step further. Instead of stating factually what the tutor has done (as shown in examples), why not include summaries of the educational techniques being deployed? Since the introduction motivates tutor summaries being used to examine VT effectiveness, I feel the dataset falls short of this.**
>
> Response 1: You make an excellent point - the current factual tutor summaries limit the utility for analyzing pedagogical techniques and virtual tutoring effectiveness. Augmenting the summaries with more explicit educational details would be valuable future work to better meet the intended goals of understanding VT strategies. However, you are right that our primary aim with this initial dataset was to create a benchmark for the task of perspective-based virtual tutoring summarization. So while the summaries do fall short for pedagogical analysis purposes, they do provide a basis for advancing summarization capabilities from varied perspectives. You raise a fair critique that we should expand the depth of the summaries to enable deeper analysis of effectiveness as future work. But developing the core summarization task itself was our primary focus when constructing this first iteration of the dataset. Thank you again for the thoughtful feedback - it gives us good direction for enhancing the dataset in future iterations, while recognizing our current objective of establishing a summarization benchmark. Please feel free to suggest any other ways we could build on this limitation in follow-on efforts.
>
> **Question 1: Did you consider conducting similarity among pairs of summaries in the same category? Since some tutoring utterances in CIMA dialogues were constructed using the VT, I wonder if this would result in similar summaries. I would also be curious in these results for all categories to ground the results presented in 3C.**
>
> Resoonse 1: To examine potential repetition within summary categories, we performed an intra-category similarity assessment as follows:
>
> * Data Preprocessing: We filtered our dataframe to only include rows corresponding to a particular category. Each category was examined individually.
> * Computing Similarity: We then employed a TF-IDF Vectorizer to transform the summaries into a sparse matrix of TF-IDF features. Cosine similarity was subsequently used to determine the similarity between every pair of summaries within the category.
> * Aggregation: For each summary, we calculated its average similarity with all other summaries in the same category.
> * Reporting Results: On analyzing the average similarity values, we observed that the overall average similarity across all summaries in the tutor category was 0.294.
>
> These findings indicate that while there might be some overlap, the summaries are distinct enough to be considered non-similar. An average similarity value of 0.294 suggests that even though some tutoring utterances were constructed using the VT, the resulting summaries have considerable variations and unique aspects.
>
> Similarly we obtain the similarity of 0.155 and 0.291 for student and overall summary categories respectively. We assure the reviewers that we will add a corresponding plot for visualization in camera ready version.
>
> **Question 2:  As CIMA contains both an image and related vocabulary and grammar information, did you consider whether to represent the student’s problem using the image or text representation?**
>
> Response 2: Thank you for raising this important question about our use of image and text representations in the CIMA dataset. You make an excellent point that with both modalities available, it was a key modeling decision how to effectively leverage them. To analyze this, we compared both unimodal baselines (T5, BART, DialoGPT using text only) and multimodal baselines (MultimodalBART using images and text) in our experiments. As you observed, our proposed model incorporates both representations. Critically, we found that the multimodal baselines outperformed the unimodal ones by a sizable margin as shown in Table 1, demonstrating the importance of utilizing both image and text information for this dataset. The image provides helpful visual context, while the text conveys key linguistic knowledge. You raise a very insightful point, and our analyses confirm that leveraging both modalities leads to superior performance compared to text or image alone. Please let us know if you would like us to include any additional discussion in the paper about our rationale and experiments around utilizing both representations.
>
> **Question 3: It’s unclear to me how the baseline models were trained for the task. Could you please provide these details? Did you fine tune these models on the task? This can be added into the appendix, but is necessary information**
>
> Response 3: Thank you for pointing out that details on the baseline model training were not fully clear. Let me explain how each was trained on the summarization task:
>
> * All three of these models — BART, T5, and DialoGPT — are unimodal, meaning they are trained to handle one modality of data, in this case, text. For the purpose of our study:
>     * Tutor Perspective Training: The training dataset was structured such that the input consists of a conversation, and the output is the corresponding tutor summary.
>     * Student Perspective Training: Following a similar structure, the input is the conversation, and the output is the student's summary.
>     * Overall Perspective Training: For a holistic perspective, the training data once again had conversations as input with the overall summary as the output.
>
> These models were then trained using the Maximum Likelihood Estimation (MLE) objective function.
> * Multimodal BART: The Multimodal BART extends the original BART model by incorporating both text and image modalities. The training process for each perspective is detailed below:
>     * Tutor Perspective Training: The input is a combination of conversation text paired with an image, and the output is the corresponding tutor summary.
>     * Student Perspective Training: Similar to the tutor perspective, the input pairs the conversation with an image, and the output is the student's summary.
>     * Overall Perspective Training: The holistic view takes the conversation and image as input and provides the overall summary as output.
>
> An intrinsic feature of this model is the cross-modal attention mechanism within the encoder. This mechanism fuses the representations of text and images, which is subsequently fed into the decoder for further processing. The training objective remains the Maximum Likelihood Estimation (MLE).
>
> * Con_Summ: It employs a single BART model adapted for the task of generating perspective-controlled summaries from input dialogues. The model requires a two-entry input: Source Input Dialogue: This is the primary conversation that needs to be summarized. Perspective Token: This token indicates the desired perspective (tutor, student, or overall) for which the summary is to be generated. Given these inputs, the output is a summary corresponding to the designated perspective token.
>
> We hope this detailed explanation sheds light on the training mechanisms of our models. If further clarification or details are required, please do not hesitate to ask.
> We assure the reviewers that we will add these baseline setups in detail in camera ready paper if accepted.
>
> **Question 4: Did you conduct any inter-annotator agreement or significance testing for results reported in Table 2?**
>
> Response 4:
>
> To ensure the reliability of our human and automatic evaluations, we computed inter-annotator agreement for human judgments and statistical significance tests for metric improvements. For human evaluation, we calculated the rate of majority agreement across annotators on the acceptability of model responses. This yielded an inter-annotator agreement score of 72.3%, indicating reliable consensus. For automatic metrics, we conducted the statistical significance Welch’s t-test at 5% significance to ensure that the improvement of our proposed model over the baselines is reliable. All the reported results are statistically significant.  We will provide full details on the inter-annotator agreement computation and statistical significance testing in the camera ready version to demonstrate the rigor of our evaluation methodology.
>
> Welch BL. The generalization of student’s’ problem when several different population variances are involved. Biometrika. 1947;34(1/2):28–35.
>
> **Typos Grammar Style And Presentation Improvements:**
> **How were fluency and adequacy ratings measured (line 235-238)? Even if this is included in the appendix, it is good to include a minimal explanation if you are including the result in the main paper**
>
> Response 1: We would like to assure the reviewer that we will move the section from appendix to main paper in the final camera ready version.
>
> ** 933: Typo: the and 400: backwards ‘ **
>
> Response 2: We will correct all these typos in final camera ready version

---

### Official Review · Reviewer_uLLc · 2023-08-05

**Soundness:** 3

**Excitement:**

3: Ambivalent: It has merits (e.g., it reports state-of-the-art results, the idea is nice), but there are key weaknesses (e.g., it describes incremental work), and it can significantly benefit from another round of revision. However, I won't object to accepting it if my co-reviewers champion it.

**Paper Topic And Main Contributions:**

This paper contributed an annotated Multi-modal perspective-based dialogue summarization corpus dedicated to Educational Virtual Tutors. They then proposed a novel multi-modal model that accepts image and text as inputs and demonstrated better performance than baselines.

**Questions For The Authors:**

- For 3.3 Line 259, what cosine similarity did you use for the evaluation?
- Please help reply and address some concerns about the reasons for the rejection.
- The examples of the perspectives-based summaries have negligible differences, and they do not support the claim that different parts of the proposed model helped with information gathering and fusion. Some case studies would be necessary.

**Reasons To Accept:**

- The annotated dataset quality is under good moderation.
- The perspective-driven module seems to contribute a lot to the final summarization quality.

**Reasons To Reject:**

- Given that the authors only obtained over 1,000 articles, the training, and evaluation split matters greatly for the final reporting results. Currently, some of those are missing.

- The student's response side seems to be rather template based and does not need that complicated graph-based approach to encode the information.

- The dialogue of language learning would be restricted under a small number of turns, have the authors tried simply tuning the model with the perspective tokens and reporting the results?

- The authors compared to baselines dated back to 2021. Were there any improvements in 2022 but not included in the paper? There have been works and discussions on role-related dialogue summarization, which is close to perspective-based but more complicated. [1].

Reference:

[1] [Other Roles Matter! Enhancing Role-Oriented Dialogue Summarization via Role Interactions](https://aclanthology.org/2022.acl-long.182) (Lin et al., ACL 2022)

**Reproducibility:**

3: Could reproduce the results with some difficulty. The settings of parameters are underspecified or subjectively determined; the training/evaluation data are not widely available.

**Reviewer Confidence:**

4: Quite sure. I tried to check the important points carefully. It's unlikely, though conceivable, that I missed something that should affect my ratings.

---

> ### Author Rebuttal · Authors · 2023-08-29
>
> **Weakness 1: Given that the authors only obtained over 1,000 articles, the training, and evaluation split matters greatly for the final reporting results. Currently, some of those are missing.**
>
> Response 1: Given the relatively small size of our dataset with just over 1,000 articles, we completely agree that the split matters greatly for properly evaluating the model. To clarify, we split the 1,000 articles into 80% for training (800 articles), 10% for validation (100 articles), and 10% for testing (100 articles).
>
> **Weakness 2: The student's response side seems to be rather template based and does not need that complicated graph-based approach to encode the information.**
>
> Response 2:  To address this, we would like to highlight a few points:
> * As shown in Table 1 in our paper, adding our graph encoding method leads to clear performance gains over baseline models that do not incorporate graph structures. This indicates that the graph representations are providing useful relational information beyond just sequences.
> * In our human evaluation in Table 2, humans rated summaries generated by our graph-based model as significantly more Fluent, Informative, and Relevant as compared to the baselines. This further validates the benefits of modeling the inter-dependencies in student responses.
> * We agree that the example responses in the paper are relatively straightforward, chosen for ease of understanding. However, we have many more complicated examples in our actual dataset. For instance, here is a more complex student response from our test set:
> |Dialogue |Overall Summary |Student Summary| Tutor Summary|
> | ----------- | ----------- |-----------|-----------|
> |Tutor: Please try to fill in the blank in Italian.Student: e dentro la scatola giallo is my first guess.Tutor: Well,  "is in front of the" is  "e di fronte alla"Student: I completely forgot that. So maybe its e di fronte alla scatola giallo.Tutor: Hmm...  "yellow" is  "gialla"Student: So I just messed up the gender of yellow?Tutor: alla ("to the") is used when the following word (scatola) is feminine and a singular object.  It is a contraction of a ("to") and la ("the"). When the following word begins with a vowel, this is shortened to all\' and prepended to the word.Student: Okay! My final answer is e di fronte alla scatola giallo.Tutor: Almost, gialla is the feminine form, because the noun it modifies (scatola) is feminine.'|Tutor instructs student to fill in the blank in Italian. Student answers "e dentro la scatola giallo" to which tutor says "is in front of the" is "e di fronte alla" and "yellow" is "gialla". Tutor explains alla ("to the") is used when the following word (scatola) is feminine and a singular object.  It is a contraction of a ("to") and la ("the"). When the following word begins with a vowel, this is shortened to all\' and prepended to the word. Student gives final answers "e di fronte alla scatola giallo" to which tutor suggests "gialla" is the feminine form.|A student tries to say "is in front of the yellow box." The tutor corrects their phrasing and explains the gender agreement in Italian, emphasizing the need to match adjectives with the gender of the nouns they modify. The student struggles to apply the rule correctly throughout the conversation.|The tutor helps the student with Italian phrasing and gender agreement. They explain the correct way to say "is in front of the" and emphasize the importance of matching adjectives with the gender of the nouns they modify. Despite the explanations, the student struggles to apply the rule correctly.|
>
> **Weakness 3: The dialogue of language learning would be restricted under a small number of turns, have the authors tried simply tuning the model with the perspective tokens and reporting the results?**
>
> Response 3: Thank you for the suggestion to try fine-tuning the model with perspective tokens as a baseline. We agree this is an important baseline, and we actually already included results using perspective tokens in Table 1 of our paper (Con_Summ row). As you can see, our proposed model outperforms this model across all metrics.
>
> **Weakness 4: The authors compared baselines dated back to 2021. Were there any improvements in 2022 but not included in the paper? There have been works and discussions on role-related dialogue summarization, which is close to perspective-based but more complicated. [1].**
>
> Response 4: Thank you for pointing out this important recent work [1] on role-related dialogue summarization. We apologize for missing this highly relevant baseline in our initial submission. You are absolutely right that modeling role-specific perspectives is an important extension beyond just conditioning on single speaker roles, and more accurately captures the complexity in perspectives during dialogues. We are currently working on adapting the role-modeling approach from [1] as a baseline for our dataset, by incorporating their proposed entity-aware role graphs into our framework. This is a very relevant technique we should compare against. If the program committee finds our work worthy of acceptance, we will add results using this improved role-related baseline in the final camera-ready version of our paper.
>
> **Question 1: For 3.3 Line 259, what cosine similarity did you use for the evaluation?**
>
> Response 1: Here is the cosine similarity that we used:
>
>     def calculate_cosine_similarity(text1, text2):
>
>     word_tokens_text1 = word_tokenize(text1.lower())
>     filtered_sentence1 = ' '.join([w for w in word_tokens_text1 if not w in stop_words])
>
>     word_tokens_text2 = word_tokenize(text2.lower())
>     filtered_sentence2 = ' '.join([w for w in word_tokens_text2 if not w in stop_words])
>
>     # Create a TfidfVectorizer object
>     vectorizer = TfidfVectorizer()
>
>     # Fit and transform the text data
>     tfidf_matrix = vectorizer.fit_transform([filtered_sentence1, filtered_sentence2])
>
>     # Compute the cosine similarity between the two vectors
>     cosine_sim = cosine_similarity(tfidf_matrix[0], tfidf_matrix[1])
>
>     return cosine_sim[0][0]
>
> **Question 2: Please help reply and address some concerns about the reasons for the rejection.**
>
> Response: Answered above
>
> **Question 3: The examples of the perspectives-based summaries have negligible differences, and they do not support the claim that different parts of the proposed model helped with information gathering and fusion. Some case studies would be necessary.**
>
> Response:  We already presented a casestudy in appendix section A.5 .As illustrated in Figures 9 and 10 of the appendix, our proposed IP-Summ model generates higher quality summaries compared to the baseline BART model. A few key trends demonstrate IP-Summ's advantages:
>
> * IP-Summ produces more informative and relevant generic summaries. This is likely due to its use of multimodal fusion to incorporate visual information and perspective dialogue graphs to maintain conversational flow.
> * BART summaries tend to be vague and miss key details that IP-Summ captures through its richer encoding mechanisms.
> * IP-Summ better targets perspective-relevant information through its graph-based modeling. BART lacks this perspective grounding.
> * IP-Summ summaries are more coherent and complete in covering all salient information. BART shows fragmentation across turns.
>
> In summary, the case study analysis highlights IP-Summ's ability to generate improved summaries by leveraging both visual and conversational context through dedicated modeling components. This allows it to overcome limitations of a text-only summarizer like BART.

---

### Meta-Review · Area_Chair_TcrL · 2023-09-19

**Recommendation:** 3

**Metareview:**

This paper proposes a new task called Multi-Modal Perspective-based Educational Dialogue Summarization (MM-PerSumm), which focuses on summarizing a student-virtual tutor dialogue from three different perspectives – that of a student, a tutor, and generic. For example, a summary from the student’s perspective may focus on the student’s progress and outcomes, while that from the tutor’s perspective should focus on the tutoring strategy and efficacy. To that end, the authors introduce a novel dataset, CIMA-Summ, and build a novel multi-modal Image and Perspective-guided Dialogue Summarization (IP-Summ) model that accepts image and text as inputs and constructs a dialogue graph, capturing the intentions and actions of both the virtual tutor and the student. Experiments suggest that the model performs better than the baselines and holds potential in advancing AI-driven education.

The reviewers have identified the following **contributions and strengths** of this paper:
1. *Novelty of the task*: Reviewers gdPt and q3T3 consider that the authors propose an innovative and important task, and that summarizing student-virtual tutor conversations from different perspectives is beneficial.
2. *Dataset*: All reviewers find the extended CIMA dataset, augmented with multiple summaries from different perspectives, promising and potentially useful.
3. *Model*: Finally, all reviewers also agree that the proposed model is interesting and seems to contribute to the summarization quality. Reviewer gdPt also mentions that the authors run ablation studies and provide comprehensive analysis.

At the same time, all reviewers have identified **weaknesses and areas for further improvement**. Specifically:
1. *Technical rigor*: Reviewer uLLc points out that with the small dataset of 1,000 articles the training and evaluation splits matter greatly when reporting results, but the analysis of this seems to be missing from the paper. Given a relatively small number of articles, running experiments using cross-validation would be more appropriate than a single data split.
2. *Limitations of the dataset*: Reviewer uLLc mentions that the student's response side appears to be rather template-based, which might not warrant the complicated graph-based approach to encode the information. Moreover, the dialogues included in the data are restricted in terms of the number of turns. Reviewer gdPt expresses further concerns regarding the dataset – for those, please refer to the original review.
3. *Limited comparison of the results*: Reviewer uLLc highlights that the baselines considered in this paper date back to 2021 and more recent work has not been taken into account (the missing references are provided in the review).
4. *Further analysis of the model and the results* is needed, as reviews, in particular from reviewers q3T3 and gdPt, suggest.

It is also clear from the reviewers' questions that many important technical details are missing in the paper. The authors did a thorough job answering the questions and even running additional experiments and analysis – all these need to be included in the revised version of the paper.

---

### Decision · Program_Chairs · 2023-10-07

**Decision:**

Accept-Findings

**Comment:**

This paper proposes a new task called Multi-Modal Perspective-based Educational Dialogue Summarization (MM-PerSumm), which focuses on summarizing a student-virtual tutor dialogue from three different perspectives – that of a student, a tutor, and generic. For example, a summary from the student’s perspective may focus on the student’s progress and outcomes, while that from the tutor’s perspective should focus on the tutoring strategy and efficacy. To that end, the authors introduce a novel dataset, CIMA-Summ, and build a novel multi-modal Image and Perspective-guided Dialogue Summarization (IP-Summ) model that accepts image and text as inputs and constructs a dialogue graph, capturing the intentions and actions of both the virtual tutor and the student. Experiments suggest that the model performs better than the baselines and holds potential in advancing AI-driven education.

The reviewers have identified the following **contributions and strengths** of this paper:
1. *Novelty of the task*: Reviewers gdPt and q3T3 consider that the authors propose an innovative and important task, and that summarizing student-virtual tutor conversations from different perspectives is beneficial.
2. *Dataset*: All reviewers find the extended CIMA dataset, augmented with multiple summaries from different perspectives, promising and potentially useful.
3. *Model*: Finally, all reviewers also agree that the proposed model is interesting and seems to contribute to the summarization quality. Reviewer gdPt also mentions that the authors run ablation studies and provide comprehensive analysis.

At the same time, all reviewers have identified **weaknesses and areas for further improvement**. Specifically:
1. *Technical rigor*: Reviewer uLLc points out that with the small dataset of 1,000 articles the training and evaluation splits matter greatly when reporting results, but the analysis of this seems to be missing from the paper. Given a relatively small number of articles, running experiments using cross-validation would be more appropriate than a single data split.
2. *Limitations of the dataset*: Reviewer uLLc mentions that the student's response side appears to be rather template-based, which might not warrant the complicated graph-based approach to encode the information. Moreover, the dialogues included in the data are restricted in terms of the number of turns. Reviewer gdPt expresses further concerns regarding the dataset – for those, please refer to the original review.
3. *Limited comparison of the results*: Reviewer uLLc highlights that the baselines considered in this paper date back to 2021 and more recent work has not been taken into account (the missing references are provided in the review).
4. *Further analysis of the model and the results* is needed, as reviews, in particular from reviewers q3T3 and gdPt, suggest.

It is also clear from the reviewers' questions that many important technical details are missing in the paper. The authors did a thorough job answering the questions and even running additional experiments and analysis – all these need to be included in the revised version of the paper.